# BRIDGING PCA AND NEURAL NETWORKS: NEW INSIGHTS INTO CLASS BIAS

## ABSTRACT

Understanding class-level hardness is essential for addressing class bias in machine learning. Traditionally, class bias has been explored with two primary approaches: analyzing raw input data to improve preprocessing strategies or examining neural network latent representations to refine model training. In this work, we find that PCA-transformed spaces—despite being produced through linear transformations—still contain substantial information about class-level hardness. This suggests that, despite their distinct goals and methodologies, both PCA and neural networks may encode similar features related to class bias, offering new insights into the nature of class bias and how data representations are formed in both PCA and neural networks.

Analyzing class bias commonly involves Pearson Correlation, which assumes stable inputs. However, we find that class bias is a highly unstable phenomenon with respect to variables such as training time and model initialization, with class-level variability often exceeding the differences between classes. Together with increased variability in class accuracies over dataset-level ones, this suggests that current methods for addressing dataset-level variability may be inadequate for handling class bias.

## 1 INTRODUCTION

Class bias, where class-level hardness is not uniform across classes, is a significant phenomenon in machine learning. This issue has traditionally been explored in the context of imbalanced datasets, where classes are represented by unequal numbers of samples (Kubat et al., 1997); often where the minority classes is of critical importance such as healthcare diagnostics (Fotouhi et al., 2019) or fraud detection (Makki et al., 2019). These imbalances naturally lead to models favouring majority class over the minority class. However, imbalance ratio is just one of the factors responsible for class bias. In fact, recent studies challenge the assumption that imbalance ratios are the primary cause of class bias (Sinha et al., 2020; 2022), revealing that this bias persists even in balanced datasets (Ma et al., 2023b). This insight allows studying class bias in scenarios where imbalance ratios are not a confounding factor.

Investigation of class bias can be performed principally at two stages: the input space and the latent space. In the input space, we scrutinize raw data to discern which classes are likely to be harder to learn. This approach facilitates more effective data preprocessing and reduces the risk of harm from uniform data augmentation (Balestriero et al., 2022; Kirichenko et al., 2023). In the latent space, we focus on the impact of various latent representations on class performance, which allows us to devise specialized regularization techniques that guide neural networks toward better representations (Ma et al., 2023a;b).

From a geometrical standpoint (Narayanan & Mitter, 2010; Fefferman et al., 2016), inference involves disentangling class manifolds (Olah, 2014; Brahma et al., 2015). The geometric properties of these manifolds influence the difficulty of class disentangling (Griffin et al., 2024), and these properties evolve as data progresses through neural network layers (Naitzat et al., 2020; Magai, 2023; Suresh et al., 2024). As a result, metrics like curvature are ineffective at estimating class bias in early layers but become strong indicators in later layers, while the opposite trend is observed for class separation metrics (Ma et al., 2023b).

Principal Component Analysis (PCA), in contrast, is a straightforward dimensionality reduction technique that projects raw data into a lower-dimensional space using linear transformations. Thus, the representations generated by PCA differ significantly from those produced by neural networks due to their distinct objectives. In this work, we discover that PCA-transformed spaces contain valuable information about class-level hardness. For example, the curvature of class manifolds within these spaces serves as a strong indicator of class-level hardness, paralleling findings in neural network latent spaces. This discovery introduces new research directions in class-level hardness, allowing usage of more cheaply obtainable data representations when estimating class hardness, and expanding our understanding of class bias.

To effectively analyze class hardness, our approach necessitates categorizing data samples rather than treating hardness as a continuous spectrum. During this process, we found that when we apply various metrics to the data, the resulting distributions of many metrics follow inverse cumulative distribution function reminiscent to that of a Gaussian distribution. This discovery suggests that a three-tier categorization into easy, medium, and hard samples is more representative than the commonly used binary split into easy and hard (Sorscher et al., 2022; Seedat et al., 2024). This approach is particularly important for other tasks that require explicit categorization, such as data pruning (Sorscher et al., 2022), where this more granular categorization can lead to better performance.

To assess which properties of class manifolds most influence class hardness, researchers commonly employ Pearson correlation (Ma et al., 2023a;b; 2024; Kaushik et al., 2024). However, this approach assumes that class bias is a consistent phenomenon with minimal variation across model runs. Contrary to this assumption, our results show that the variability in class-level performance resulting from different random initializations can be greater than the differences observed between classes themselves. This leads to inconsistent rankings of class hardness based solely on initialization, highlighting a significant limitation of using correlation-based approaches for understanding class bias. Our findings suggest that existing methods to account for dataset-level variability might not be robust enough for addressing the nuances of class bias. In summary our work makes the following contributions:

1. **Estimating class bias using PCA-transformed spaces:** We find that PCA-transformed spaces can effectively estimate class bias, similar to those obtained from neural network latent representations. This discovery opens new avenues for research on class bias and enhances our understanding of latent representations in machine learning.

2. **Unravelling complexity of evaluating class-level hardness:** We show that variations in class-level performance across ensemble models is often larger than the differences in performance between classes themselves. Coupled with higher variability in class-level performance over dataset-level, this highlights new challenges in addressing class bias and suggests current methods may lack robustness.

3. **Refining categorization of hardness:** Our results suggest that categorizing samples into easy, medium and hard is a closer representation of the underlying hardness distributions than the commonly used easy and hard categorization. This finding can be significant for fields that by definition cannot consider hardness as a spectrum, and rely on hardness categorization, such as data pruning.

## 2 BACKGROUND

Evolution of intrinsic dimension (ID) across network layers was explored by Ansuini et al. (2019), showing that ID initially increases before decreasing in the final layers, and found that the ID of the latent dataset manifold in the last hidden layer reliably indicates dataset-level hardness. Extending this, Pope et al. (2021) demonstrated a correlation between the average ID of randomly sampled class pairs in the input space and dataset-level hardness, suggesting that ID remains a valuable metric for estimating class bias even after model bias is incorporated into data representations. Building on these findings, Ma et al. (2024) further examined the relationship between ID and model fairness, highlighting the broader utility of ID for understanding model behavior.

Naitzat et al. (2020) demonstrated that the topology of class manifolds simplifies through the layers of a well-trained network, as measured by decreasing Betti numbers. Extending this, Magai (2023) observed that changes in the persistent homological fractal dimension differ across architectures

based on convolutions versus attention mechanisms. Building on these insights, Suresh et al. (2024) proposed using the rate of decay in topological complexity, defined as the sum of Betti numbers, to quantify the effect of architectural choices on dataset-level hardness.

Kaufman & Azencot (2023) found that trained CNNs exhibit a characteristic curvature profile—an initial steep increase, followed by a plateau, and then another rise—and demonstrated that the curvature gap between the last two layers strongly correlates with dataset-level hardness. Meanwhile, Ma et al. (2023b) showed that the average curvature of latent class manifolds is a reliable indicator of class-level hardness, while curvature in the input space is not.

Ma et al. (2023a) found a strong positive correlation between the volumes of the learned embeddings (latent manifolds) and class bias, demonstrating that this correlation holds both with and without accounting for inter-class distances. Similarly, Kaushik et al. (2024) demonstrated that the spectrum, defined as the set of eigenvalues of the covariance matrix of latent manifolds, is an effective indicator of class-level hardness. Both studies highlight that class dispersion within latent space correlates with class-level hardness.

Kienitz et al. (2022) found that dataset entanglement, estimated through the entanglement of representative pair of similar classes using a Linear Support Vector Classifier, provides a better estimate of dataset-level hardness than intrinsic dimension. Additionally, Ma et al. (2023b) showed that the average separation degree between class manifolds is a good indicator of class-level hardness in the input space but not in latent space.

Hardness has been extensively studied at the dataset level, leading to numerous metrics, as outlined in several surveys (Ho & Basu, 2000; Ho et al., 2006; Barella et al., 2021) and supported by open-source libraries (Komorniczak & Ksieniewicz, 2023; Lorena et al., 2024). While most of these metrics focus on dataset-level complexity (Smith et al., 2014), Lorena et al. (2019) suggests that majority can be adapted for instance-level hardness.

## 3 INVESTIGATING DATA-BASED HARDNESS IDENTIFIERS

**Hardness identifiers** In this work we use fourteen different instance-level metrics that measure: 1) intra-class structure (DCC, ADSC, MDSC); 2) separation from other classes (N3, DNOC, ADOC, MDOC, CP); 3) inter-class comparison (CDR, MDR, ADR); and 5) curvature (MC, GC). However, when modified to perform as class-level hardness identifiers, the role these metrics changes. Below we categorize the instance-level metrics we used based on their role as class-level hardness identifiers:

1. **Class Dispersion:** Distance to class centroids (**DCC**).

2. **Class Density:** Average (**ADSC**), and minimum (**MDSC**) distance to same-class samples within kNN.

3. **Class Separation and Overlap:** The adapted N3 metric (**N3**) (Gøttcke et al., 2023), which checks if the nearest neighbor belongs to another class, along with distances to other-class centroids (**DNOC**), the average (**ADOC**) and minimum (**MDOC**) distances to other-class samples within the 40NN, and class purity (**CP**) within the 40NN (Xiong et al., 2012). Apart from that, Ratios of within- to between-class distances for centroids (**CDR**), 40NN minimum distances (**MDR**), and 40NN average distances (**ADR**)

4. **Geometric properties:** Mean curvature (**MC**) and Gaussian curvature (**GC**), as curvature has been shown to correlate with hardness (Kienitz et al., 2022; Kaufman & Azencot, 2023; Ma et al., 2023b).

For class dispersion we add three class-level metrics: 1) algorithm from Ma et al. (2023a) (**V**); and 2) maximum (**max** $\lambda$), and average (**avg** $\lambda$) of eigenvalues of the covariance matrix, used by Kaushik et al. (2024). In Appendix A, we provide precise information on how each of these metrics is computed. We use forty for $k$ in kNN, based on Ma et al. (2023b) who used it to measure the curvature of latent manifolds. We also use models-based methods, such as Cleanlab (Northcutt et al., 2021), EL2N Paul et al. (2021) and margin (Pleiss et al., 2020) to behave as baseline.

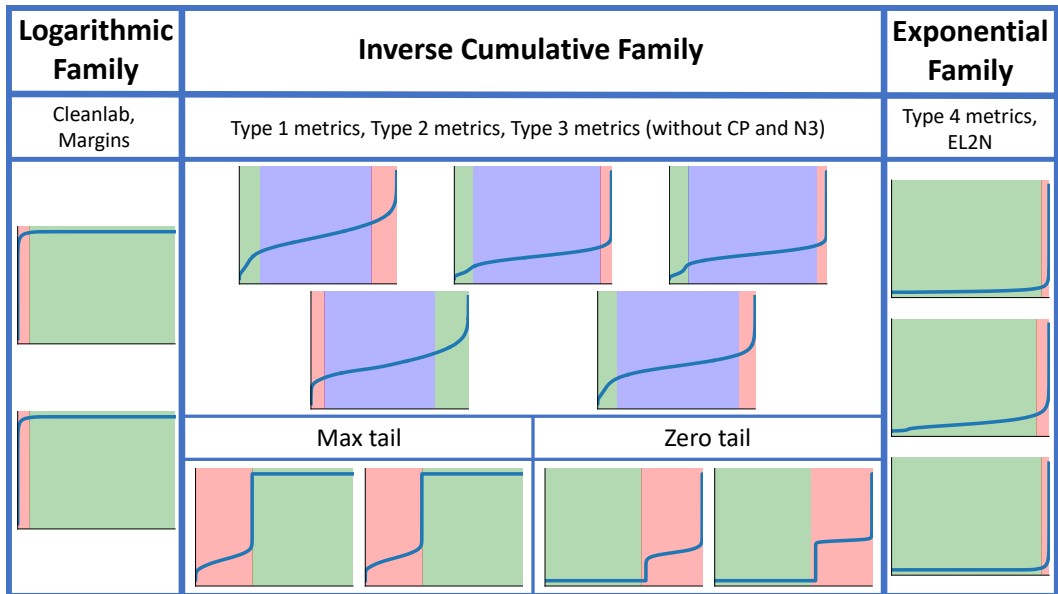

Figure 1: Classification of hardness identifiers into three families based on the distribution patterns of their metric values. The sorted data indices are divided into easy (green), medium (blue), and hard (red) categories using adaptive division points, identified by analyzing the gradients of the distribution functions. Due to neighborhood heterogeneity issues in 40NN metrics, we observe the emergence of zero and max tails, leading to the classification of some metrics from the second family into the logarithmic and exponential families, respectively. The fact that majority of metrics belong to Inverse Cumulative Family implies that categorizing data samples into easy, medium, and hard is more natural than the commonly used easy and hard. The above was obtained in *full* setting on MNIST. This Figure does not include N3 and CP.

**Experimental design** We apply our metrics to MNIST, KMNIST, FashionMNIST, and CIFAR10 under two scenarios: *full* and *part* information. In the *full* scenario, metrics are computed on the entire dataset, similar to curriculum learning, active learning, or data pruning. Here, we measure hardness based on all of the available data samples. On the other hand, in *part* scenario we apply metrics only to the training set. This setting is introduced to assess how data-based methods, many of which rely on kNN, are affected by incomplete access to information.

In order to compute the class bias, which is essential when working with class-level hardness, we train ensembles of networks for each dataset. On CIFAR10, we use ResNet56 (He et al., 2016), training 25 networks for hundred epochs with Adam (Kingma & Ba, 2015) (lr=0.01, weight decay=$1e-4$, cosine scheduler). For MNIST, KMNIST, and FashionMNIST, we use LeNet, training hundred networks for ten epochs with SGD (lr=0.001). All experiments use a batch size of 32.

**Analysing distributions of metric values** After computing the metric values for each datum, we sort the data samples based on these values. This reveals three families of metrics, based on how their values are distributed: 1) logarithmic; 2) inverse cumulative; and 3) exponential. As shown in Fig. 1, the majority of metrics belong to the second family. Notably, even when the setting or dataset changes, most metrics consistently remain in the same family. The only exceptions are N3 and Purity, which switch from the logarithmic to the exponential family depending on the dataset. While the first and third families support easy and hard sample divisions, the second family introduces a medium-hardness category.

For most hardness identifiers, high values correspond to hard samples, though there are some exceptions. Specifically, data samples with low values of DCC, MDSC, and ADSC are considered hard. A high DCC value is a simple identifier of OOD data, while large MDSC and ADSC values indicate that a sample lies in a region of low density. Similarly, Cleanlab assigns low values to samples that are likely mislabeled, and a low margin suggests that the model lacks confidence in

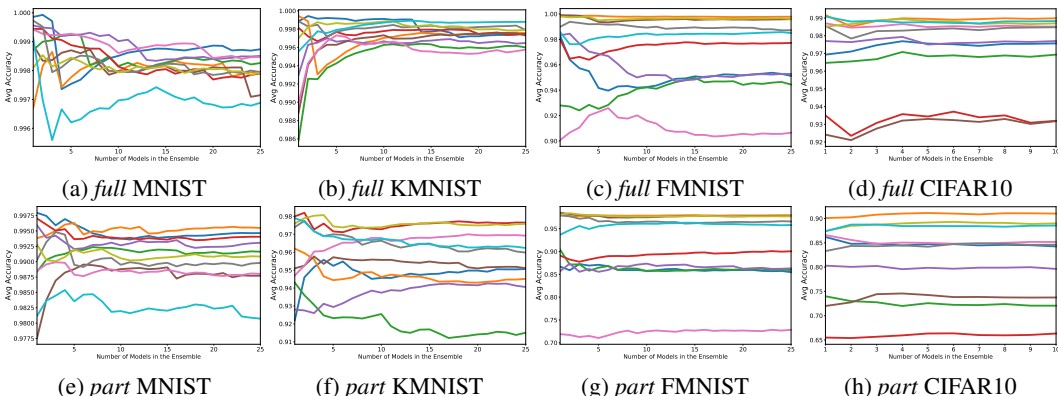

Figure 2: Class bias on MNIST, KMNIST, FashionMNIST, and CIFAR10 as we increase the number of models in an ensemble, with each colored line corresponding to separate class. First and second rows show result in *full*, and *part* information setting, respectively. We find that for MNIST and KMNIST the variations of average accuracies are larger than inter-class differences if ensemble is not large enough. We also notice that the order of class complexities in *full* setting is not the same as in *part* setting showcasing differences between approximation and generalization error.

its predictions—both of which also signify hard samples. For the remaining metrics, high values consistently indicate hard samples.

In some cases, the 40NN neighborhood for a sample contains only samples from the same class or only from other classes, causing certain metrics to return None. For Type 1 metrics, which measure the distance to same-class samples, None occurs when no same-class neighbors are found in the 40NN, indicating a hard-to-learn sample. We replace None with infinity to reflect this difficulty. In contrast, Type 2 metrics, which measure the distance to other-class samples, return None when no other-class neighbors are present, indicating the sample is easy to learn. In this case, we replace None with zero. These replacements result in distinct distribution tails: long maximum tails for Type 1 metrics (classified into the logarithmic family) and long zero tails for Type 2 metrics (classified into the exponential family).

Due to varying gradient dynamics across metrics, fixed division points can distort difficulty classification. To address this, we adopt adaptive division points to classify samples as easy, medium, or hard based on gradient values. Since most metrics yield an inverse cumulative distribution similar to that of a Gaussian distribution, we set the division points where the gradient consistently falls below the bottom $2.5\%$ of the range between the maximum and minimum gradient values. This corresponds to approximately $\pm2$ standard deviations, capturing the most extreme easy and hard samples, while the middle region reflects samples with moderate difficulty. For the first (logarithmic) and third (exponential) families, we also use the $2.5\%$ threshold to identify the end of the plateau regions, ensuring adaptive categorization of samples based on where the gradient behavior changes

## 4 COMPUTING AND INVESTIGATING CLASS BIAS

Class hardness is is evidenced by the presence of class bias. Hence, to measure the performance of metrics as class-level hardness identifiers it is essential to firstly compute class bias. Furthermore, it's important to differentiate between hardness emerging due to different types of errors. Hence, we consider two settings: *full* and *part*. In the *full* setting, we train on the entire dataset and measure class bias based on accuracies across the entire dataset, providing insights into classes that are difficult to learn due to approximation error. In the *part* setting, we train on the training set and evaluate class bias based on test set accuracies, which accounts for both approximation and generalization errors. As far as we know, this is the first work making such distinction when working with class-level hardness.

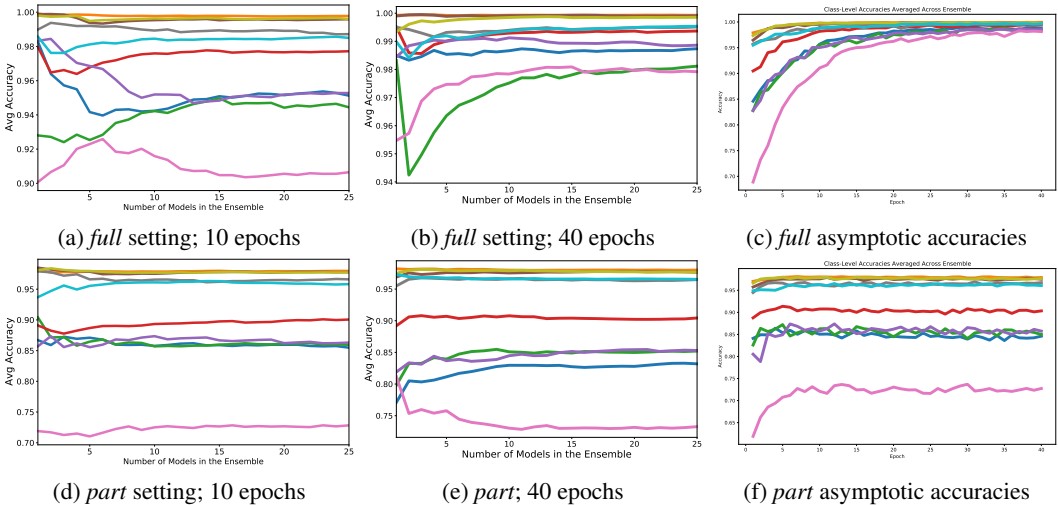

(a) *full* setting; 10 epochs      (b) *full* setting; 40 epochs      (c) *full* asymptotic accuracies

(d) *part* setting; 10 epochs      (e) *part*; 40 epochs      (f) *part* asymptotic accuracies

Figure 3: Impact of increasing the power of an ensemble, by training for thirty more epochs, on the observed class bias, together with average asymptotic accuracies of an ensemble.

**Inconsistency of class bias**    We first measure the average class-level accuracies as a function of the number of models in an ensemble. Intuitively, this function should plateau after a few iterations, with high fluctuations indicating inconsistent class-level performance across models. Since the effectiveness of metrics as class-level hardness identifiers is commonly assessed using Pearson correlation (Ma et al., 2023a;b; 2024; Kaushik et al., 2024), it is essential to establish the consistency of class bias. However, our results demonstrate that in MNIST, KMNIST and FashionMNIST *class bias is often inconsistent, with variations in class-level accuracies surpassing intra-class differences* (see Fig. 2). This inconsistency makes Pearson correlation extremely poor measure for evaluating class-level hardness metrics, as the results depend heavily on the size of the ensemble. Notably, we observe this phenomenon primarily in simpler datasets, with KMNIST and MNIST showing the most severe effects, while it is negligible in CIFAR10. This raises an important question: is the inconsistency of class bias a data or model related issue. In other words, as class-level performance improves and the differences between classes shrink, will the class-level variations also decrease proportionally, ensuring stable class bias?

To investigate this, we perform preliminary experiment where we train another ensembles on FashionMNIST using forty epochs 3. We find that for *full* setting, the differences in the performance among classes reduces significantly with the increase of strength of an ensemble. This results in more fluctuating or unstable measurements of class bias, resembling those observed in MNIST and KMNIST in Fig. 2. Meanwhile, in the *part* setting we observe minimal effects of increased ensemble power on the class-level performance and variability. This most likely happens due to limited generalization capabilities of LeNet, and we might observe similar trends to these observed in *full* setting when using more powerful model. When analyzing the evolution of the average asymptotic class-level accuracies of the ensembles (see Fig. 3c and 3f) the variability of class bias becomes even more apparent. These results clearly suggest, that the order of classes based on hardness is not fixed, and varies depending on multum of factors. To summarize, our observations suggest that the inconsistency of class bias is primarily a model-driven phenomenon, rather than being an inherent property of the data itself.

**Intricacies of class-level hardness**    Our experiments also reveal that the final ranking of class difficulties differs between the *part* and *full* setting. For example, in CIFAR10 we notice that the class represented by the brown line is the most difficult in *full* setting (Fig 2d), but only the third most difficult in *part* setting (Fig 2h). This discrepancy arises from differences in approximation and generalization errors—some difficult samples may be easier to approximate but harder to generalize on, and vice versa. This highlights that *class bias should be distinguished based on whether it results from approximation or generalization difficulties*.

Table 1: Comparing class- and dataset-level accuracies of models within an ensemble shows that the class-level variations are significantly larger than dataset-level ones. This highlights more intricate nature of class-level hardness.

| Dataset | Min Class Std | Max Class Std | Avg Class Std | Dataset Std |
|---|---|---|---|---|
| *full* MNIST | 0.0013 | 0.0050 | 0.0021 | 0.0008 |
| *full* KMNIST | 0.0010 | 0.0044 | 0.0029 | 0.0011 |
| *full* FashionMNIST | 0.0014 | 0.0264 | 0.0129 | 0.0037 |
| *full* CIFAR-10 | 0.0033 | 0.0172 | 0.0077 | 0.0056 |
| *part* MNIST | 0.0027 | 0.0097 | 0.0040 | 0.0016 |
| *part* KMNIST | 0.0087 | 0.0237 | 0.0133 | 0.0043 |
| *part* FashionMNIST | 0.0043 | 0.0371 | 0.0164 | 0.0031 |
| *part* CIFAR-10 | 0.0087 | 0.0174 | 0.0120 | 0.0032 |

We also find that the average standard deviation of class-level accuracies is larger than the standard deviation of dataset-level accuracies across models within an ensemble (see Table 1). This is most notable on FashionMNIST, where the standard deviation of dataset-level accuracies of an ensemble is equal to $0.0037$, while the average standard deviation of class-level accuracies of the same models is over three times higher, with the standard deviation of one class going over $0.025$. This discrepancy is a natural consequence of averaging, as dataset-level accuracy is a weighted average of class-level accuracies. However, it highlights that class-level hardness is inherently more complex than dataset-level hardness, due to larger standard deviations. While it is well known that model performance varies based on initialization, and ensembles mitigate this at the dataset level, the problem becomes magnified at the class level. The increased standard deviations emphasize that *simple measures used to ensure invariability in dataset-level settings may not be sufficient for class-level analyses.*

## 5 CLASS-LEVEL PERFORMANCE VS INSTANCE-LEVEL PERFORMANCE

After computing class bias, the next step is to evaluate the class-level performance of various hardness identifiers. This is done by measuring the correlation between class-level hardness identifiers and class bias. To the best of our knowledge, Ma et al. (2023b) is the only study that uses instance-level metrics as class-level hardness identifiers, transforming them by averaging instance-level values to obtain class-level metrics. However, this method discards valuable information inherent in the instance-level granularity of the metrics, potentially overlooking important insights related to class-level hardness. As an alternative, we propose to analyze the distribution of hard and easy samples within each class, rather than relying on averages. This allows us to capture the extremes—how many easy or hard samples are concentrated in a given class—and compare this distribution to the observed class bias. In our experiments we use Spearman's rank correlation. We analyze the class-level distribution of these samples and compare it to class bias, expecting positive correlation for easy samples and negative correlation for hard samples.

**Performance of class-level metrics obtained from raw data** Our results reaffirm prior research, showing that curvature is a poor class-level hardness identifier, while class separation is a strong one when measured from raw data. In fact, we discover that for MNIST, KMNIST, and FashionMNIST, even straightforward metrics like the distance to other-class samples provide a reliable estimate of class hardness. However, for more complex datasets like CIFAR10, more advanced techniques, such as the *class separation* proposed by Ma et al. (2023b), become essential for properly capturing inter-class relationships. On the other hand, both type dispersion- and density-based metrics appear to be weak class-level hardness identifiers. While the results vary by dataset, even in the best cases, they are only marginally statistically significant, as indicated by p-values below 0.05 but above 0.1. We also find that both the full and part settings yield similar results, confirming that data- and model-based methods remain effective even when reducing the amount of information $14\% - 16\%$.

**Investigating impact of PCA** To assess the impact of PCA-based dimensionality reduction on hardness identifiers, we halve the dimensionality of each dataset and repeat the process of metric

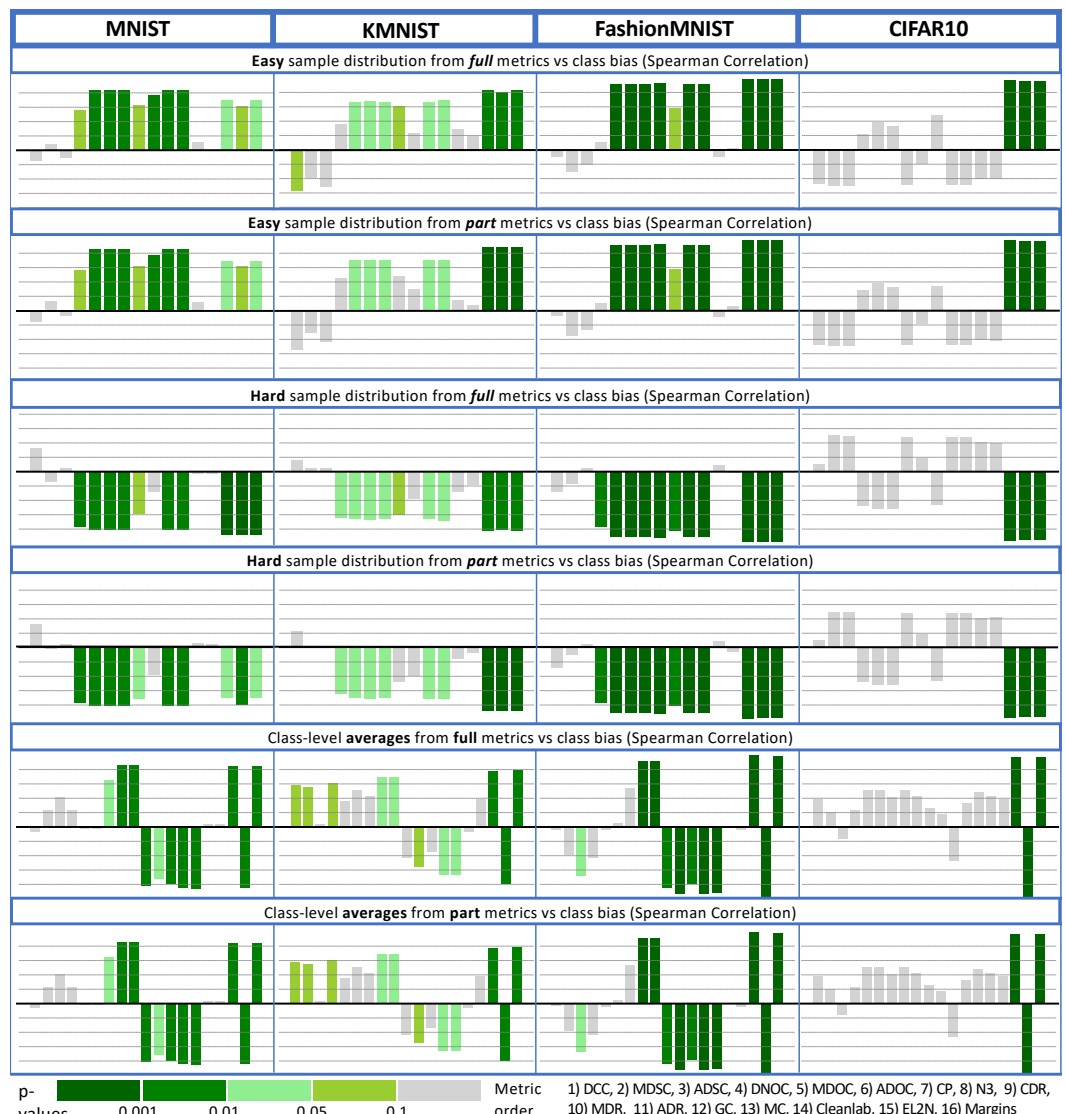

Figure 4: Performance of various metrics, obtained from raw data, as class-level hardness identifiers . Bar height indicates Spearman correlation values, with horizontal lines marking intervals of 0.2 on the y-axis. Bar color represents p-values. For class-level averages, the results also include three volume estimates: V, max $\lambda$, and avg $\lambda$, shown as the leftmost bars.

application, data partitioning, and correlation measurement, using the transformed data instead of the raw data.

Our analysis reveals that the correlations obtained after applying PCA (Fig. 5) differ significantly from those using raw data (Fig. 4). When applied to raw data, curvature was not a statistically significant metric, as indicated by p-values exceeding 0.1. However, curvature becomes a strong class-level hardness identifier in PCA-transformed spaces, with a PCC around 0.8 and p-values below 0.05, sometimes below 0.001. Meanwhile, many separation-based metrics experience a reduction in effectiveness. These patterns align with the observations of Ma et al. (2023b) conducted using metrics applied to latent spaces of deep neural networks (DNNs). This suggests that PCA and DNNs infuse similar biases into data representations. Conversely, dispersion-based techniques remain poor hardness identifiers in our study, which contrasts with Ma et al. (2023a)'s findings where volume computed from neural network representations was effective. Interestingly, we also find that in the experiments with PCA-transformed spaces, the performance of metrics is significantly more

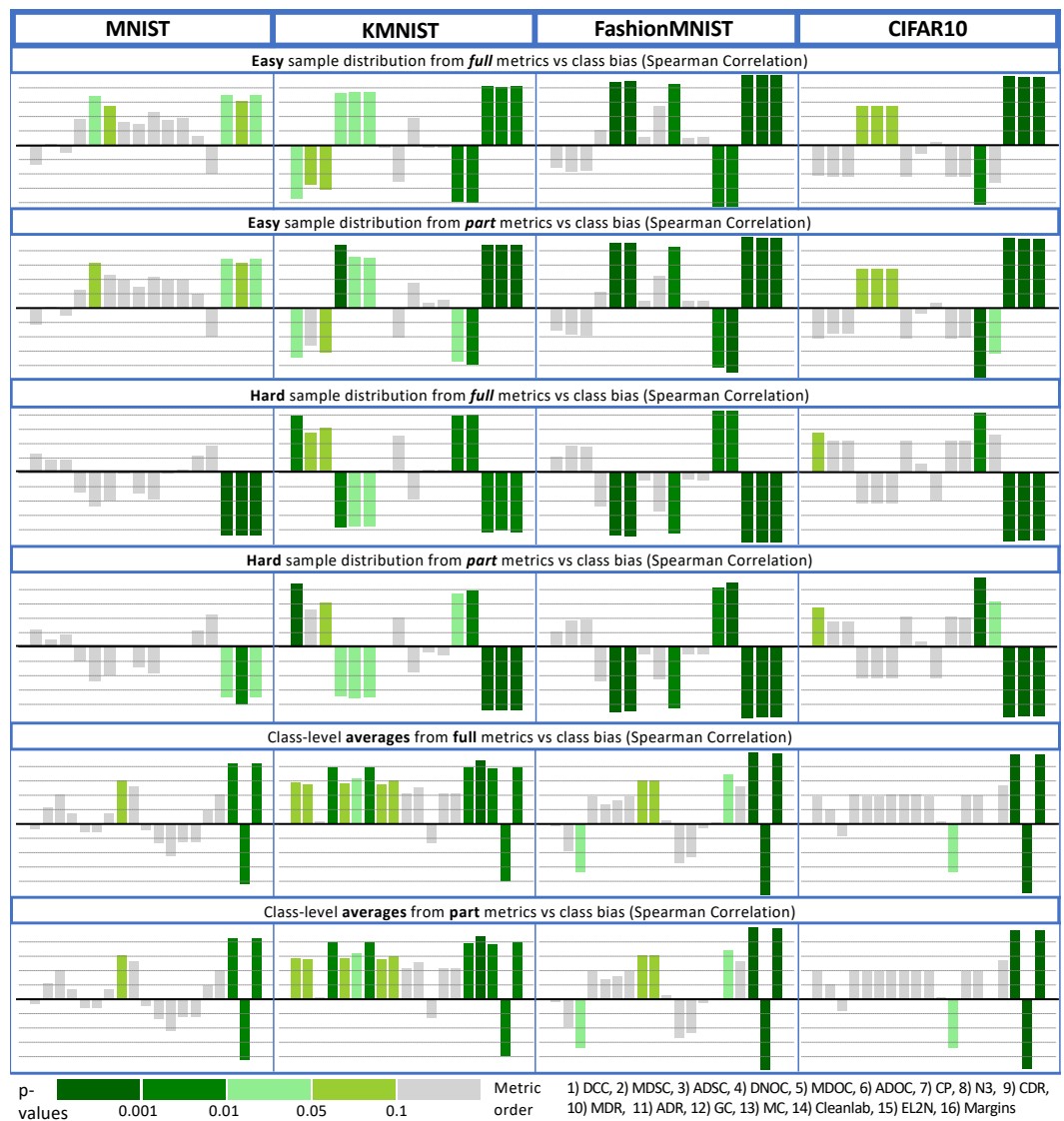

Figure 5: Performance of various metrics as class-level hardness identifiers, obtained from data representations after reducing the dimensionality of the datasets by half using PCA.

dataset-dependent than in our experiments involving raw data. Contrary to that, literature suggest dataset-consistent behaviour of class-level hardness identifiers applied to latent representations of DNNs. A promising future direction is to understand what dataset properties enable PCA-computed representations to reveal valuable information on class bias.

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

# A    DETAILED DESCRIPTION OF USED HARDNESS IDENTIFIERS

In this section, we provide a detailed explanation of the hardness identifiers used in our experiments, including their mathematical formulation and the abbreviations we refer to in the main paper.

**Notation** Let $x \in \mathbb{R}^d$ denote a data sample, and $y$ be the label of $x$. Let $\text{kNN}(x)$ represent the $k$-nearest neighbors of $x$ from the dataset, and $x' \in \text{kNN}(x)$ be one of these neighbors with label $y'$. We define the class centroid $C_y$ as the mean of all samples from class $y$.

## A.1    TYPE 1: CLASS DISPERSION

**Distance to Class Centroid (DCC)**    This metric computes the Euclidean distance between a sample $x$ and the centroid $C_y$ of its class $y$. The centroid is computed as the mean of all samples in the same class.

$$\text{DCC}(x) = \|x - C_y\|$$

When averaged over all samples in a class, the average DCC measures how far, on average, class members are from the class centroid. This reflects the dispersion or compactness of the class. A lower average DCC indicates that samples are tightly clustered around the centroid, suggesting higher class density.

## A.2    TYPE 2: CLASS DENSITY

**Minimum Distance to Same-Class Neighbors (MDSC)**    This metric computes the minimum distance between a sample $x$ and the subset of its $k$-nearest neighbors that belong to class $y$.

$$\text{MDSC}(x) = \min \{\|x - x'\| \, : \, x' \in \text{kNN}(x), \, y' = y\}$$

**Average Distance to Same-Class Neighbors (ADSC)**    This metric computes the average distance between a sample $x$ and the subset of its $k$-nearest neighbors that belong to class $y$.

$$\text{ADSC}(x) = \frac{1}{|\{x' \in \text{kNN}(x) : y' = y\}|} \sum_{x' \in \text{kNN}(x), \, y' = y} \|x - x'\|$$

Averaging these metrics at the class level provides insight into the average proximity of class members to one another within their local neighborhoods. Lower values indicate that samples are closer to their same-class neighbors, signifying higher local density within the class.

## A.3    TYPE 3: CLASS SEPARATION AND OVERLAP

These metrics measure how far a sample is from other classes, providing insights into inter-class separability.

**Distance to Nearest Other-Class Centroid (DNOC)**    This metric computes the Euclidean distance between a sample and the closest centroid of other class.

$$\text{DNOC}(x) = \min_{C_{y'} \neq C_y} \|x - C_{y'}\|$$

**Minimum Distance to Other-Class Neighbors (MDOC)**  This metric calculates the minimum distance between a sample $x$ and the subset of its $k$-nearest neighbors that belong to classes other than $y$.

$$\text{MDOC}(x) = \min \left\{ \|x - x'\| \; : \; x' \in \text{kNN}(x), \; y' \neq y \right\}$$

**Average Distance to Other-Class Neighbors (ADOC)**  This metric calculates the average distance between a sample $x$ and the subset of its $k$-nearest neighbors that belong to classes other than $y$.

$$\text{ADOC}(x) = \frac{1}{|\{x' \in \text{kNN}(x) : y' \neq y\}|} \sum_{x' \in \text{kNN}(x), \, y' \neq y} \|x - x'\|$$

**Adapted N3 (N3)**  This metric checks if the nearest neighbor of a sample comes from a different class. It returns 1 if the nearest neighbor belongs to another class, and 0 otherwise.

$$\text{N3}(x) = \begin{cases} 1 & \text{if } \text{NN}(x) \neq y \\ 0 & \text{if } \text{NN}(x) = y \end{cases}$$

**kNN Class Purity (CP)**  This metric computes the proportion of $k$-nearest neighbors from classes other than $y$ within the $k$-nearest neighbors, indicating how mixed the neighborhood is.

$$\text{CP}(x) = \frac{|\{x' \in \text{kNN}(x) : y' \neq y\}|}{k}$$

**Centroid Distance Ratio (CDR)**  This metric calculates the ratio between the distance to the same-class centroid and the distance to the nearest other-class centroid.

$$\text{CDR}(x) = \frac{\text{DCC}(x)}{\text{DNOC}(x)}$$

**Minimum Distance Ratio (MDR)**  This metric computes the ratio between the minimum distance to same-class neighbor and the minimum distance to other-class neighbor.

$$\text{MDR}(x) = \frac{\text{MDSC}(x)}{\text{MDOC}(x)}$$

**Average Distance Ratio (ADR)**  This metric calculates the ratio between the average distance to same-class neighbors and the average distance to other-class neighbors.

$$\text{ADR}(x) = \frac{\text{ADSC}(x)}{\text{ADOC}(x)}$$

### A.4   TYPE 4: GEOMETRIC PROPERTIES

For the curvature-based metrics, we use the code and algorithms developed by Ma et al. to compute the **Mean Curvature (MC)** and **Gaussian Curvature (GC)** of the data manifold. These metrics capture the geometric complexity around each sample, which correlates with sample hardness. For more details on the curvature estimation process, we refer the reader to Ma et al. (Ma et al., 2023b).

## B   COMPLETE RESULTS ON DISTRIBUTION OF METRIC VALUES

Show only results on MNIST and CIFAR10; for results on other datasets refer to GitHub

