# OpenReview forum: "Bridging PCA and Neural Networks: New Insights into Class Bias"
_ICLR.cc/2025/Conference — ICLR 2025 Conference Withdrawn Submission_

### Official Review · Reviewer_ahhB · 2024-10-28

**Soundness:** 2
**Presentation:** 1
**Contribution:** 3
**Rating:** 3
**Confidence:** 3

**Summary:**

This work investigates class bias, which the authors define as as the observation that a neural networks have variation in the performance on classes in classification tasks. Specifically, the authors focus on the class-balanced regime where class bias is not confounding with relative number of samples. The authors compute 14 datapoint/instance-level metrics that measure class hardness. They show that These metrics can be separated into three families depending upon the distribution patterns (fig 1); for a subset of the fourteen metrics, this suggests a 'medium-difficulty' region of sample hardness.

Having defined the metrics, they apply these to investigate relatively small vision models on common benchmark datasets with no class imbalance. The authors separate quantifying class hardness into two regimes called full and part, where full is trained and evaluated on the full dataset, and partial is trained on train and evaluated on test. This is done to separate out generalization vs approximation errors.

Having computed and making the observations presented in figure 1 re. medium difficulty, they continue to present observations on the effects of ensemble learning on class bias (fig 2), then investigating the effect of additional training (which the authors refer to as both power and strength of the ensemble) on class bias fig (3). There are two primary reasons I see this as important to the work: 1) it establishes the phenomena of class bias being present in the datasets and models used in this work. 2) it allows the authors to comment that the full and part results are different**, so they may argue that their methodology of separating full and part is useful; it is especially apparent that these are important when considering training for long times, as the authors claim that class bias variation among models in the ensemble increases in the long training time regime on FMNIST.

The authors continue to analyze the results of their experiments, observing that metrics computed from their ensemble at the dataset level belie more intricate details about the ensemble's performance at the individual class level (Table 1).

The authors then return to their hardness metrics. The authors argue that to evaluate the effectiveness of their hardness metrics, they should use a Spearman rank correlation, and measure the correlation between class-level hardness identifiers and class bias on all samples within a class(?). They show that the first 3 of their class-hardness measures are not well correlated to class-level hardness, nor are the curvature-measures. They show that class hardness metrics 5-11 (class separation and overlap) and 14-16 (model based methods developed in prior works) seem routinely correlated across each of the smaller model experiments (MNIST, KMNIST, and FMNIST) but only 14-16 seem effective on the experiment with CIFAR10.

Using PCA, the authors reduce the dimensionality of each dataset by half and recompute the class hardness metrics. (I believe the authors must use a tabular model after performing PCA instead of using a CNN on the PCA transformed space?) They observe that in doing-so they recover the results that 14-16 are effective class-bias indicators across all models, datasets, and full/part, but the other metrics are more mixed on whether they are well correlated. Their results at times support prior research or contrasts with it.

**Strengths:**

The authors make contributions to advancing methodology in two concrete ways. I do think the authors are correct to point out that full/part are two different kinds of experiments when understanding class-bias. I also understand why using Spearman correlation would be an improvement over using the Pearson correlation, given the distribution of hardness indicators.

While I have questions about various aspects, I think ostensibly visualizing the distribution of the various available instance level hardness features and categorizing them is of value to the community.

I do applaud the authors for conducting very original research and having high aims.

**Weaknesses:**

Overall I find the presentation lacking to the point of detracting from the substance of the work. I believe the article is unfocused and splits its attention too much between ideas. The paper is nebulous where instead it should be specific. While there are some good points you make I would argue you need to refocus onto just those few good points  with more targeted experiments then cut away the fat.

I'm not sure how PCA was used in the experiments-- I'm pretty sure you must have transformed the datasets using PCA and trained a tabular model, but all details about what kind of model was trained and for how long are left unstated. This is important, since you have shown that model and optimizer choice can affect the class-bias indicator correlation strength. I believe the authors intend to show that using PCA-transformed spaces they recover behavior on class level bias that parallels an original network. I think comparing the statements made in their contribution statement vs the observations, their contribution is overstated. They observe a complex relationship between class-bias metrics in original networks vs PCA transformed spaces, with only some of the features remaining effective.

regarding contribution 2: "Unraveling complexity": I think that is a lazy catch-all: the authors comment on various observations from their experiments. I do not think the authors unravel complexity, but instead demonstrate that there is a lot of complexity in class bias, which should not be a surprise. The impact of the paper would be improved if the authors designed more targeted experiments to study the relationship between controlled variables and class-bias.

The plots have multiple flaws that detract from your presentation. Figures 4 and 5 are supposed to be a contrast that firmly establishes how PCA can be used to help understand class-bias in neural networks. If you are going to focus your title and abstract so heavily on PCA (which I think is an odd choice!), I would encourage you to find a way to combine these plots so that you can make the contrast/similarities stand out to the reader.

You do not show error bars in Figures 2 and 3 computed from the ensemble of models, but then comment on the significance of your results. Without errors I'm not sure I follow exactly what experiment was conducted to create figures 2.

Minor flaws like font sizes varying between subplots, font sizes being absurdly small, and figure 3 having only one of the plots titled. I would suggest adding a vertical line to Table 1 to separate out Dataset std from the remaining columns, since the main text only uses the contrast of Dataset std being smaller compared to the other rows. Anything you can do to highlight that in the table would help the reader.

Minor: The impact of the paper is undercut by the fact that you show model choice, dataset complexity, optimizer choice, affect the correlation between class-hardness and indicators.

**Questions:**

I do wonder whether easy vs medium vs hard should be applied in the context of the hardness feature distribution shape or the actual observed hardness of the instance? Since part of the work is evaluating these various class-bias features it seems like it begs the question of whether your easy-vs-medium-vs-hard sample categorization actually bears out in your experiments?

One piece of information regarding this that is missing from figure 1 and I think not considered by the work is how each of these features are correlated with one another? specifically, what is the correlation matrix of class-bias hardness measures for these experiments?

Finally, there is a line re. where some hardness indicators are incomputable you replace None with infinity (lines 242-3). Can you explain whether this choice affects your visualization in Figure 1, the computation of adaptive division points, and your Spearman correlation.

---

### Official Review · Reviewer_xvaE · 2024-11-03

**Soundness:** 1
**Presentation:** 1
**Contribution:** 1
**Rating:** 1
**Confidence:** 4

**Summary:**

This paper is about class difficulty/hardness and its relationship with class biases, in particular about some new insights about using PCA to estimate hardness, comparisons of metrics and show many of them have small diferences that are smaller than dataset-level differences, and a proposal to use three categories of hardness (easy, medium, and hard) instead of the usual two.

The paper seems to be a comparison of methods with some new insights extracted from them, contributions are the use of PCA to estimate hardness, the complexity of evaluating class hardness, and the new categorization of hardness.

**Strengths:**

I have trouble finding strengths in this paper as there are many weaknesses. In my opinion the only strength is that the topic of data hardness and class bias is underexplored and is very interesting on its own, as data points have different difficulties that affect machine learning models and even humans, so overall the topic is very interesting and has plenty of room for contributions.

**Weaknesses:**

- The paper overall is written in a very dense way, it is hard to read and understand, mostly due to writing and trying to say too much in many sections (For example, Sec 4 in Inconsistency of class bias, results are reported and then new research questions formulated, which should be made in the conclusion), and the presentation is not good, there are plots without axis labels (like Figures 4 and 5), and the paper does not have a conclusions section, and it is still under the 10 page limit, so there was space to provide a closing section and expand on important details. Overall the paper does not look polished enough for a conference submission.
- The overall goal of this paper is not clear, this seem to be an experimental paper about some new remarks on class bias and hardness, without actually defining these terms, and at points I believe these terms are mixed up.
- The experiments are very hard to interpret because the experimental setup is not completely defined, the authors rush to results without having a clear goal, specially since this is a data-driven experimental paper. I believe the paper would have to be completely rewritten since there are three different goals (corresponding to the contributions and main sections of the paper) that are not completely related to each other. Another problem with experimental setup is about the use of pearson correlation, without clearly specifying correlation between what variables. Finally, the hypothesis for p-values in Sec 5 is not mentioned at all, and p-values enter the evaluation without a clear definition of what test was made and what reasoning led to it.
- The title of the paper is usually about the main point of the paper, and in this case it is about PCA and neural networks and class bias, but this seem to be a minor result in the experiments.
- I am not convinced about the PCA experiments, the authors repeat one experiment after applying PCA to reduce the data to half its original dimensions, and make conclusions from this, but the choice of using PCA to reduce data dimensions by half is basically arbitrary, there is no justification on why half is a sensible choice, and the whole results and conclusions might just be that enough data variance is kept by using half of the dimensions. A more sensible experiment would make experiments with a variable number of principal components being kept, or at least justify the choice of number of dimensions being kept.
- The use of easy/medium/hard and comparison using ensembles to determine hardness is very similar to some concepts in the paper "Difficulty Estimation With Action Scores for Computer Vision Tasks" by Arriaga et al 2023. In particular I believe hardness defined as easy/medium/hard is not novel as it has been presented in the literature before, but authors do not acknowledge this. Additionally the observations presented in Sec 4 about class bias being model depending are already made in the previously mentioned paper.

**Questions:**

- What is the novelty of this paper? Overall for me it is hard to see how this paper contributes to the state of the art, as there are several directions that could be independent papers.
- How was the half number of data dimensions chosen? And why are experiments not evaluated as a function of number of data dimensions kept?
- What are the hypotheses for which p-values are computed in Sec 5? This is not properly described in the paper.
- How does this paper compare to Arriaga et al. 2023? Some of the conclusions in this paper seem to be already present in the literature I just mentioned.

---

### Official Review · Reviewer_ZwCJ · 2024-11-03

**Soundness:** 2
**Presentation:** 2
**Contribution:** 2
**Rating:** 3
**Confidence:** 5

**Summary:**

The authors investigate the relationship between Principal Component Analysis (PCA) and neural networks in understanding class bias in machine learning. The authors demonstrate that PCA-transformed spaces, despite their linear nature, retain information about class-level hardness similar to neural network latent representations, suggesting that PCA encodes features related to class bias. Additionally, the study reveals that class bias is highly unstable across different training runs and model initializations, often exhibiting variability that surpasses inter-class differences. The authors propose a more nuanced categorization of sample hardness into easy, medium, and hard, rather than the traditional binary classification, which can enhance tasks like data pruning.

**Strengths:**

The paper presents an original approach by showing that PCA-transformed spaces can capture class-level hardness, offering a new perspective on class bias analysis. The experimental setup considers benchmark datasets such as MNIST and CIFAR10, and employs a diverse set of fourteen hardness metrics to validate the findings. The categorization of sample hardness into easy, medium, and hard based on metric distributions provides a nuanced framework that could improve tasks like data pruning. Additionally, the investigation into the variability of class bias across different model initializations highlights important challenges in achieving consistent class performance.

**Weaknesses:**

Unfortunately, the paper has several weaknesses that undermine its overall impact. Notably, it lacks a dedicated conclusion that summarizes findings and outlines future research directions. Instead, the authors include only a single sentence at the end of Section 5 to address future work, which is insufficient for providing a comprehensive closure to the study. The paper also claims to bridge PCA with neural networks in capturing class bias but fails to provide a direct comparison between PCA-transformed spaces and existing neural network latent representations. While the authors state that both methods "may encode similar features related to class bias", this similarity is not investigated $-$ neural networks are not part of this study. Moreover, the authors proposed categorization of samples into "easy", "medium", and "hard" is qualititative (based on graphical trends pertaining to the inverse cumulative family), which necessitates rigorous theoretical and/or empirical backing. No such backing is provided for the proposed categorization; the "improved performance in tasks such as pruning" is not empirically demonstrated. These issues collectively detract from the paper’s robustness and its ability to convincingly advance the understanding of class bias in machine learning.

**Questions:**

1. The paper claims to bridge PCA with neural network latent spaces in capturing class bias but does not provide a direct comparison. Can the authors comment on this?

2. Can the authors provide more rigorous backing (theoretical and/or empirical) for the categorization of samples into "easy", "medium" and "hard"?

---

### Official Review · Reviewer_6mFy · 2024-11-04

**Soundness:** 2
**Presentation:** 3
**Contribution:** 1
**Rating:** 3
**Confidence:** 3

**Summary:**

The paper investigates class bias in Neural Network, finding that PCA-transformed spaces retain meaningful information on class hardness, similar to neural network latent spaces. It challenges the consistency of class bias, showing high variability across model runs and suggesting that current methods may inadequately address this issue. The authors also propose a refined hardness categorization ("easy, medium, hard") over traditional binary classifications and  highlight that class bias stems more from model-driven factors than inherent data properties.

**Strengths:**

- Understanding class-bias is an important topic critical in several application

- The paper is well-written and easy to follow.

**Weaknesses:**

W1: In my perspective, The contributions of this paper is limited. major part of the analysis done here does not bring new insights on class bias. Indeed, it has been established in the literature that class bias is not consistent and that is not a data-only problem.

W2: Most of the analysis done here are using small datasets (mnist/FashionMNIST/CIFAR10) which makes any insights/analysis shown here irrelevant w.r.to the current state-of-the-art deep learning methods used nowadays. So i think the contribution (in the current state) is not beneficial for the research community.

W3:The link between NN class-bias and PCA is weak. In fact, as PCA is a linear transformation and do not have any non-linearity on the input. So, it is not surprising at all that is conserves all the class-bias coming from the data itself. However, it is known that class-bias in NN depend on several other (more puzzling/interesting) training-related (and model-related) factors (data augmentation/training optimizer, regularization technique etc). However, this can not be captured by PCA as it does not have access to the DL training.

**Questions:**

Q1: What are the practical values of the insights provided in this paper to the current state-of-the-art DL methods ?

  > Line 300-303: *This raises an important question: is the consistency of class bias a data or model related issue. In other words, as class-level performance improves and the differences between classes shrink, will the class-level variations also decrease
proportionally, ensuring stable class bias?*

Q2:  This question is not new. In fact, it has been thorough investigated in the literature (e.g., [1-3]) and it was shown that for DL, many factors influence  this class-variations, including data augmentation, regularization, label noise etc...


>  (line:267-269) In the part setting, we train on the training set and evaluate class bias based on test set accuracies, which accounts for both approximation and generalization errors. As far as we know, this is the first work making such distinction when working with class-level hardness.

Q:  the claim to be the first to study class-generalization errors is not true. See [3]

Q: have the authors tried any other linear transformation other than PCA to show that this behavior is specific to PCA?

[1] Balestriero, Randall, Leon Bottou, and Yann LeCun. "The effects of regularization and data augmentation are class dependent." Advances in Neural Information Processing Systems 35 (2022) \
[2] Kirichenko, Polina, et al. "Understanding the class-specific effects of data augmentations." ICLR 2023 Workshop on Pitfalls of limited data and computation for Trustworthy ML. 2023.\
[3] Laakom, Firas, Yuheng Bu, and Moncef Gabbouj. "Class-wise Generalization Error: an Information-Theoretic Analysis." arXiv preprint arXiv:2401.02904 (2024).

---

### Note · Authors · 2024-11-25

**Comment:**

We thank the reviewers for their time and constructive feedback. We will incorporate these suggestions into the revised version of our paper.

**Withdrawal Confirmation:**

I have read and agree with the venue's withdrawal policy on behalf of myself and my co-authors.